# The Emerging Role of PET/CT with PSMA-Targeting Radiopharmaceuticals in Clear Cell Renal Cancer: An Updated Systematic Review

**DOI:** 10.3390/cancers15020355

**Published:** 2023-01-05

**Authors:** Alessio Rizzo, Manuela Racca, Sara Dall’Armellina, Pasquale Rescigno, Giuseppe Luigi Banna, Domenico Albano, Francesco Dondi, Francesco Bertagna, Salvatore Annunziata, Giorgio Treglia

**Affiliations:** 1Department of Nuclear Medicine, Candiolo Cancer Institute, FPO–IRCCS, 10060 Turin, Italy; 2Nuclear Medicine Unit, Department of Medical Sciences, AOU Città della Salute e della Scienza, University of Turin, 10126 Turin, Italy; 3Department of Oncology, Candiolo Cancer Institute, FPO-IRCCS, 10060 Turin, Italy; 4Portsmouth Hospitals University NHS Trust, Portsmouth PO12UP, UK; 5Division of Nuclear Medicine, Università degli Studi di Brescia and ASST Spedali Civili di Brescia, 25123 Brescia, Italy; 6Unità di Medicina Nucleare, TracerGLab, Dipartimento di Diagnostica per Immagini, Radioterapia Oncologica ed Ematologia, Fondazione Policlinico Universitario A. Gemelli, IRCCS, 00168 Rome, Italy; 7Clinic of Nuclear Medicine, Imaging Institute of Southern Switzerland, Ente Ospedaliero Cantonale, 6501 Bellinzona, Switzerland; 8Faculty of Biology and Medicine, University of Lausanne, 1011 Lausanne, Switzerland; 9Faculty of Biomedical Sciences, Università della Svizzera Italiana, 6900 Lugano, Switzerland

**Keywords:** PSMA, PET, nuclear medicine, kidney, clear cell renal cell carcinoma, oncology, imaging

## Abstract

**Simple Summary:**

Positron emission tomography/computed tomography (PET/CT) with prostate-specific membrane antigen (PSMA)-targeting radiopharmaceuticals is a relatively novel technique currently employed in the management of prostate cancer patients. These radiopharmaceuticals target PSMA, also known as carboxypeptidase type II, which is mainly expressed by prostate cancer cells. Nevertheless, its expression has been observed in the neovasculature of various kinds of cancer, including clear cell renal cell carcinoma (ccRCC). Based on this biological mechanism, several authors postulated a potential role for this diagnostic method in ccRCC patients in different clinical settings. This systematic review provides an overview of the possible applications of PET/CT with PSMA-targeting radiopharmaceuticals in ccRCC.

**Abstract:**

Background: Recent articles proposed the employment of positron emission tomography/computed tomography (PET/CT) with prostate-specific membrane antigen (PSMA)-targeting radiopharmaceuticals in clear cell renal cell carcinoma (ccRCC). Methods: The authors performed a comprehensive literature search of studies on the performance of PET/CT with PSMA-targeting radiopharmaceuticals in ccRCC. Original articles concerning this imaging examination were included in newly diagnosed ccRCC patients and ccRCC patients with disease recurrence. Results: A total of sixteen papers concerning the diagnostic performance of PSMA-targeted PET/CT in ccRCC (331 patients) were included in this systematic review. The included articles demonstrated an excellent detection rate of PSMA-targeting PET/CT in ccRCC. Conclusions: PSMA-targeted PET/CT seems promising in detecting ccRCC lesions as well as in discriminating the presence of aggressive phenotypes. Prospective multicentric studies are warranted to strengthen the role of PSMA-targeting PET/CT in ccRCC.

## 1. Introduction

Kidney cancer is a common kind of tumor with an estimated incidence of approximately 400,000 new diagnoses every year worldwide [1,2]. Renal cell carcinoma (RCC) is the most common type of renal tumor [3]. Based on its histopathological features, we divide RCC into two types: clear cell RCC (ccRCC) and non-ccRCC (which categorizes at least 15 histotypes, of which the most frequent consist of papillary RCC and chromophobe RCC) [4]. However, the morphologic classification of RCC is going to become obsolete since new genomic and molecular features of these malignancies are emerging [5]. Most of our knowledge about the genetic basis of RCC comes from studies concerning its inherited forms, including von Hippel–Lindau (VHL) (ccRCC), hereditary papillary RCC (type I papillary RCC), and hereditary leiomyomatosis RCC (type II papillary RCC) [6,7]. There is a clear relationship between the prevalence of kidney cancer and VHL genotype, with a higher prevalence in patients with partial deletion of the VHL gene [8]. In contrast, no VHL gene mutations were found in patients with papillary, chromophobe, collecting duct, or medullary kidney cancers [9,10,11,12].

Due to the tumor cells’ overproduction of platelet-derived growth factor (PDGF) and vascular endothelial growth factor (VEGF), ccRCC is considered a highly vascularized tumor [4].

To date, most ccRCC diagnoses result from incidental findings as a consequence of the widespread use of noninvasive radiological techniques, including ultrasonography (US), magnetic resonance imaging (MRI), or computed tomography (CT), performed for another reason [11]. Moreover, paraneoplastic syndromes’ symptoms caused by hormones or cytokines excreted by tumor cells or by an immune response against the tumor are not uncommon in RCC, accounting for hypercalcemia, fever, and erythrocytosis [12]. Imaging examinations usually strongly suggest the diagnosis, although ccRCCs can display variable radiographic appearances [13]. Typical radiological ccRCC characteristics include exophytic growth, heterogeneity due to intratumoral necrosis or hemorrhage, and high contrast enhancement in CT and MRI examinations [14]. In current clinical practice, total-body contrast-enhanced CT is essential for optimal staging and restaging of ccRCC since it allows us to assess the primary tumor’s size and extension, lymph node involvement, and to evaluate the presence of distant metastases. MRI can also provide additional information to determine whether the tumor extends into the vasculature. Moreover, the development of radiological techniques and the availability of data derived from the Human Genome Project introduced the concept of radiogenomics. Radiogenomics, which can be applied to RCC, consists of studying the association of imaging features of a specific disease and its gene expression patterns or molecular phenotype. Its purpose is to noninvasively collect additional data for diagnosis, staging, prognostication, and the assessment of an optimal therapy based on the imaging features of the disease [15].

Nuclear medicine examinations allow physicians to detect functional abnormalities that might anticipate morphological findings in conventional imaging in oncological malignancies. With regard to nuclear medicine imaging, the most studied examination in ccRCC patients is fluorine-18 fluorodeoxyglucose ([^18^F]FDG) positron emission tomography (PET)/CT [16]. [^18^F]FDG PET/CT is not currently recommended as an imaging method in ccRCC by practice guidelines [17,18,19] and does not play a key role in managing ccRCC patients so far for several reasons. The first issue concerns the excretion of [^18^F]FDG and its metabolites, seeing as how their physiological renal excretion hinders the characterization of primary lesions, making the differentiation from the physiologic background challenging [16]. Furthermore, [^18^F]FDG PET/CT is not routinely employed to evaluate ccRCC because of its relatively low [^18^F]FDG uptake; the underlying mechanism for this phenomenon is partially unclear yet. Nevertheless, a recent study showed that [^18^F]FDG uptake reflected FBP1 expression levels in patients with ccRCC since it was higher in ccRCCs with low fructose 1,6-bisphosphatase 1 (FBP1) expression compared to patients with high FBP1 expression [20].

Carboxypeptidase type II, also known as prostate-specific membrane antigen (PSMA), is a transmembrane protein encoded by the gene FOLH1 [21,22]. Several authors observed that carboxypeptidase type II could be expressed on the surface of neovascular endothelial cells of various solid tumors other than prostate cancer [23]. In this context, different radiolabeled PSMA-targeting low-molecular-weight molecules were introduced to enhance nuclear medicine imaging and to operate as theranostic agents for patients affected by metastatic prostate cancer [24,25]. Since neovascular endothelium cells of various malignancies, including ccRCC, overexpress PSMA, and since ccRCC is considered a highly vascularized tumor, this might be a rationale to employ PET imaging with PSMA-targeting radiopharmaceuticals in tumors with variable [^18^F]FDG uptake, exploring different biological processes than glucose metabolism. Furthermore, since a combination of immunotherapy and antineoangiogenetic therapy are the standard of care in metastatic ccRCC [17,18,19], PSMA-targeted PET/CT might be a valuable tool to anticipate the treatment’s outcome and to assess the response to therapies in ccRCC patients.

Various recent studies evaluated the performance of PSMA-targeted PET/CT in ccRCC. This paper has the purpose of accomplishing an updated systematic review concerning PSMA-targeted PET/CT performance in patients affected by ccRCC. Furthermore, this paper aims to gather evidence to compare the diagnostic performance of PET with PSMA-radioligands to other instrumental examinations in ccRCC.

## 2. Materials and Methods

### 2.1. Protocol

This systematic review was conducted following a preconceived protocol [26] since the authors employed the “Preferred Reporting Items for a Systematic Review and Meta-Analysis” (PRISMA 2020 statement) as a touchstone in its completion [27]. The PRISMA checklist is accessible as Appendix A. This systematic review was not preregistered in any electronic database.

Firstly, a review query was established: can PSMA-targeting PET/CT help to detect ccRCC lesions?

In agreement with the Population, Intervention, Comparator, Outcomes (PICO) framework, two reviewers (G.T. and A.R.) accomplished a literature search, establishing criteria for the eligibility of the studies found in the literature search: patients with ccRCC diagnosis (Population) submitted to PET with PSMA-targeting radiopharmaceuticals (Intervention) compared or not to traditional imaging or [^18^F]FDG PET/CT (Comparator); the predetermined outcomes were the evaluation of PSMA-targeting radiopharmaceuticals uptake in ccRCC and PSMA-targeted PET detection rate in ccRCC.

The same authors independently carried out the comprehensive literature search, the election of the papers, and the assessment of their quality. Any discrepancy between the reviewers was solved through an online consensus meeting.

### 2.2. Literature Search Strategy and Information Sources

Once the review query was defined, two authors (G.T. and A.R.) independently fulfilled a literature search using two electronic bibliographic databases (Cochrane library and PubMed/MEDLINE), seeking studies assessing the diagnostic accuracy of PSMA-targeted PET in ccRCC.

The following search algorithm was employed: (A) “PSMA” AND (B) “clear cell “ OR “kidney” OR “renal” OR “ccRCC” OR “RCC” AND (C) “PET” OR “positron.” The authors did not apply any restriction concerning the publication date and language. Moreover, the authors screened the bibliography of the included studies searching for additional suitable articles to improve the research. Finally, ongoing studies were evaluated through ClinicalTrials.gov database.

The literature search was lastly updated on 24 October 2022.

### 2.3. Eligibility Criteria

Original papers reporting data about the use of PSMA-targeted PET in different clinical settings (characterization of ccRCC lesions, staging and restaging of ccRCC patients) were eligible for inclusion in this systematic review. Letters, comments, editorials, reviews concerning the selected topic, original papers regarding different fields of interest (including preclinical studies or studies not using PSMA-targeted PET in ccRCC), and case reports or small case series concerning the analyzed topic were excluded from the systematic review.

### 2.4. Selection Process

Two authors (G.T and A.R.) independently reviewed titles and abstracts of the papers obtained through the literature search and assessed if they were suitable for inclusion in the systematic review. Finally, the reviewers opted for inclusion or exclusion, stating the reason for all the screened records.

### 2.5. Data Collection Process and Data Extraction

Two authors (G.T. and A.R.) independently gathered all the included papers to avoid potential biases and extracted information, exploiting the data in the text, the figures, and the tables. For each included study, the following data were extracted: general study information (experimenter, publication date, nation, design of the experimentation, and eventual funding); patient characteristics (number of included patients, age, sex ratio, clinical setting, grading, and other instrumental examinations); index text characteristics (type of PSMA-targeting radiopharmaceutical administered, hybrid imaging protocol, patient preparation, injected activity, elapsed time between radiopharmaceutical administration and image acquisition, and image analysis), data concerning the detection rate of PSMA-targeted PET in ccRCC on a per-patient- and per-lesion-based analyses, and diagnostic reference standard.

### 2.6. Quality Assessment (Risk of Bias Assessment)

Since diagnostic accuracy studies included in systematic reviews usually report heterogeneous results due to differences in the design, the QUADAS-2, a valuable instrument to evaluate the quality of diagnostic test accuracy studies, was the selected method to assess the risk of bias and the applicability in individual studies [28]. Two reviewers (G.T and A.R) independently assessed the studies’ quality in the systematic review. Four domains (patient selection, index test, reference standard, and flow and timing) were considered concerning the risk of bias, and three fields were evaluated regarding applicability (patient selection, index test, and reference standard). Any discrepancies among the reviewers about the quality assessment were solved by online consensus.

## 3. Results

### 3.1. Literature Search and Study Selection

The literature search was lastly updated on 24 October 2022 and revealed a total of 340 records. Based on the inclusion and exclusion criteria mentioned above, 328 articles were excluded (290 as not in the field of interest, 27 as reviews, editorials, or letters, and 11 as case reports). Twelve remaining records were eligible for inclusion in the systematic review (qualitative synthesis) after a full-text assessment [29,30,31,32,33,34,35,36,37,38,39,40]. Two additional papers were assessed as suitable for inclusion after screening the references of these articles [41,42]. Figure 1 summarizes the study selection process.

### 3.2. Study Characteristics

Analyses of the characteristics of the fourteen articles eligible for the systematic review (qualitative research), including 331 ccRCC patients, are presented in Table 1, Table 2 and Table 3. With regard to the general study information (Table 1), the included papers were published between 2016 and 2022 in Europe, Asia, Australia, and the USA. Three studies had a prospective design, whereas the remaining eleven were retrospective. Among the fourteen studies included, only two were multicentric [38,39]. Only one of the revised articles declared funding in its text [42].

With regard to the patient key characteristics (Table 2), the number of ccRCC-included patients ranged from 8 to 54 patients (average age range: 57–66 years; male percentage range: 25.7–100%). PSMA-targeted PET/CT was used for staging in newly diagnosed ccRCC patients in seven studies [29,30,33,34,36,37,42], to restage ccRCC patients with suspect relapsing disease in three articles [32,35,40], and for both purposes in the remaining four articles [31,38,39,41]. Six studies classified the analyzed lesions according to the World Health Organization/International Society of Urological Pathology (WHO/ISUP) grading score [33,34,35,36,37,39], and five included an immunohistochemistry analysis to evaluate PSMA staining in the neovasculature of ccRCC lesions [33,34,35,36,42]. Concerning the comparison of PSMA-targeting PET/CT to other instrumental diagnostic examinations, four studies employed diagnostic CT [29,31,35,41], two alternated diagnostic CT and MRI [30,42], one used [^18^F]FDG PET/CT [32], one used only MRI [39], two employed both [^18^F]FDG PET/CT and diagnostic CT [38,40]; the remaining four studies did not use any comparative imaging but relied on pathology to validate their findings [33,34,36,37].

The included studies showed significant differences in the index test key characteristics (Table 3). In eight articles, the only injected tracer was [^68^Ga]Ga-PSMA-11 (activity range: 75–272 MBq in absolute values) [29,31,33,34,35,36,37,42]; in two studies, only [^18^F]F-DCFPyL was injected (mean injected activity: 333 MBq) [30,32]; in one, the only employed radiopharmaceutical was [^18^F]F-PSMA-1007 (activity range: 217–268 MBq); in two papers, an alternate between [^68^Ga]Ga-PSMA-11 and [^18^F]F-PSMA-1007 was reported based on radiopharmaceuticals availability [38,39]; and in the remaining study, both [^68^Ga]Ga-PSMA-11 and [^18^F]F-DCFPyL were used [40]. All the analyzed papers reported PET images coregistration with low-dose CT, and in one, a dynamic PET acquisition of the upper abdomen was performed [34]. Excluding the only study which conducted a dynamic acquisition [34], the uptake time after PSMA-targeting radiopharmaceutical administration and PET scan varied from 45 to 142 min. All the included articles performed qualitative and semiquantitative analyses within the PET imaging interpretation. Semiquantitative analyses were performed calculating the mean, maximal, and peak-standardized uptake values (SUV_mean_, SUV_max_, and SUV_peak_, respectively). The metabolic tumor volume (MTV) and total lesion PSMA uptake (TLP) were measured in one study [40]. The target-to-background uptake ratio (TBR) was assessed in four studies by dividing the lesions’ SUV_max_ for the background, muscle tissue, or liver SUV (SUV_max_ or SUV_mean_) [32,34,35,42]; finally, intralesional radiopharmaceutical kinetics were deepened only in one study [34].

### 3.3. Risk of Bias and Applicability

The overall assessment of the risk of bias and concerns about the applicability of the included papers according to QUADAS-2 is provided in Figure 2.

### 3.4. Results of Individual Studies (Qualitative Synthesis)

The comprehensive evaluation of PSMA-targeted PET/CT in detecting ccRCC lesions assessed an optimal diagnostic performance in all the included papers, both on a per-lesion- and per-patient-based analyses in all the evaluated clinical settings without significant differences between PET/CT scans executed for staging or restaging [31]. In this context, the reported detection rate ranged from 84% and 100% in the per-patient-based analysis [30,32,40] and from 80.5% to 100% in the per-lesion-based analysis [29,30,31,32,42], showing excellent performances in detecting metastatic lesions in lymph nodes, bone, breast, brain, pancreas, adrenal glands, liver, contralateral kidney, and lungs [29,30,35,38,40,41,42]. 

None of the included studies reported adverse effects after administering PSMA-targeting radiopharmaceuticals.

All the reviewed papers assessed variable PSMA-targeting radiopharmaceuticals uptake in primary and metastatic ccRCC lesions; most reports reported it was higher than the activity of the surrounding tissue. With regard to the semiquantitative data, average SUV_max_ reported values varied between 6.9 and 25.9 for primary lesions and ranged from 2.7 to 19.5 for metastatic lesions; the high heterogeneity observed among the included studies may be explained by the employment of three different PSMA-targeting radiopharmaceuticals in the included studies. Since every study used different background regions to calculate TBR, its variability had a poor value and was not assessed.

Most of the included studies observed that PSMA-targeting PET/CT could detect more lesions than conventional imaging, especially in metastatic lesions of the bone [29,30,31,35,38,42]. Nevertheless, in one of the included studies, PET/CT with PSMA-radioligands detected fewer lesions than diagnostic CT [40]. Finally, PSMA-targeting PET/CT showed a complementary role to MRI in detecting venous thrombi in the renal vein [39].

Two studies compared the diagnostic performances of PSMA-targeting PET/CT to [^18^F]FDG PET/CT [32,38]. Both reported a higher accuracy of the first over the latter, both in terms of number of highlighted lesions and of uptake entity, reporting higher SUV values in PSMA-targeting PET/CT.

Concerning the correlation between semiquantitative metrics at PSMA-targeted PET/CT and pathological characteristics of ccRCC, several studies observed higher PSMA-radioligands uptake in lesions with sarcomatoid or rhabdoid differentiation [29,33], whereas one study did not find significant differences in terms of uptake among lesions with or without this feature [40]. Furthermore, we found conflicting evidence about the correlation between semiquantitative values and the grade of PSMA staining in the immunohistochemistry analysis, since one paper reported a correlation between these two values [42]; conversely, another study did not find the same relationship [35]. Finally, several of the included studies deepened the association among PSMA-targeting radiopharmaceuticals uptake and WHO/ISUP grade, VEGFR-2/PDGFR-β expression, and HIF-2α expression, reporting that PSMA-targeting PET/CT might be a valuable instrument for discriminating the presence of these pathological features in ccRCC patients [33,36,37].

Regarding potential differences in the diagnostic performance of PSMA-targeted PET/CT among ccRCC and non-ccRCC tumors, several studies observed an inferior accuracy as well as a poorer uptake of PSMA-targeting radiopharmaceuticals in non-ccRCC lesions than ccRCC [31,42].

When reported, PET/CT with PSMA-radioligands could lead to a change in patients’ management (usually from local to systemic therapy) in all the evaluated clinical settings, ranging from 13% to 43.8% of the enrolled patients in eight studies [29,30,31,38,39,40,41,42].

Finally, one study explored the potential role of PSMA-targeting PET/CT to assess treatment response in ccRCC patients undergoing tyrosine kinase inhibitors (TKI) or immune checkpoint inhibitors (ICI) alongside conventional imaging [41], finding out discrepancies in 82% of the enrolled patients.

Since the studies that reported the PSMA-targeted PET/CT detection rate in ccRCC used different modalities for its assessment and different benchmarks to calculate it [29,30,31,32,40], a meta-analysis could not be accomplished. The results of the included papers, including semiquantitative metrics, detection rate, and percentage of change in patient management, are synthesized in Table 4.

## 4. Discussion

Based upon the PSMA overexpression on prostate cancer cells’ membrane, radiopharmaceuticals targeting this receptor became a relatively novel compound to improve the performance of molecular imaging and to develop new therapeutic instruments through radioligand therapy (RLT) in metastatic castration-resistant prostate cancer (although it does not substitute for laboratory examinations and MRI in its diagnosis). Furthermore, the recent literature reported that carboxypeptidase type II might be overexpressed in the neovasculature of various types of tumors other than prostate cancer, including ccRCC [43], and its expression could regulate tumor cell invasion and neoangiogenesis through the transduction of the integrin signal in the endothelium [44].

This rationale brought about several researchers to assess the diagnostic performance of PSMA-targeted PET/CT and evaluate the correlation between PSMA-targeting radiopharmaceuticals uptake and tumor histopathologic characteristics in newly diagnosed as well as previously treated ccRCC patients through the past six years [29,30,31,32,33,34,35,36,37,38,39,40,41,42]. This updated systematic review attempted to gather all the evidence concerning this emerging topic.

Despite the available evidence being quite limited and the included papers reporting different approaches to calculate the detection rate, all the authors investigating the accuracy of PET/CT with PSMA-targeting radiopharmaceuticals in ccRCC patients stated excellent performances both in per-patient- and per-lesion-based analyses in newly diagnosed patients as well as previously treated patients undergoing disease restaging without significant differences among the explored clinical settings [29,30,31,32,40].

One of the most influential prognostic factors regarding overall and progression-free survival in patients with ccRCC is the WHO/ISUP grading system based on tumor nuclear morphology, proposed by Moch et al. in 2016 [45]. In addition to this grading classification, recent shreds of evidence reported that several histopathology features, including intratumoral necrosis and the presence of rhabdoid or sarcomatoid ccRCC variants, might worsen the prognosis in ccRCC patients [46,47,48,49]. Most of the studies included in this systematic review exploring the potential of PSMA-targeting radiopharmaceuticals uptake to discriminate lesions with different WHO/ISUP grades as well as the presence of malignant histopathology features reported an SUVmax being significantly higher in ccRCC lesions with a more aggressive phenotype [29,33,40]. These results suggest that PSMA-radioligands uptake might be a valuable prognostic factor in ccRCC patients; in this context, more prospective studies are needed to assess its potential in this setting in combination with other prognostic factors as well as an independent variable.

A new diagnostic instrumental examination can change patient management once its employment brings about an up- or downstaging compared to conventional imaging and alters the previously planned treatment. Since 20–30% of patients with apparently resectable ccRCC develop metastatic disease after surgery, the current standard of care, consisting of diagnostic CT and/or MRI, might be considered insufficient [50]. When reported, PSMA-targeted PET/CT showed a superior accuracy in tumor burden characterization, highlighting more metastatic lesions than conventional imaging, reducing false positive findings, and changing patient management in a substantial percentage of cases [29,30,31,35,38,40,42]. Furthermore, in one study, PSMA-targeted PET/CT was employed to assess the presence of tumor thrombi and showed complementary results with MRI in their characterization [39]. These promising results enhance the need for more prospective clinical trials to assess how many ccRCC patients might have their treatment plan change after PSMA-targeted PET/CT and, more importantly, if its modification could actually lead to an improvement in their overall and disease-free survivals.

[^18^F]FDG PET imaging relies on the increased glucose demand by rapidly dividing tumor cells. Membrane glucose transporters, mainly GLUT-1, actively transport [^18^F]FDG into the cell, where it is converted by hexokinase into [^18^F]FDG-6-phosphate, which is trapped in the cytoplasm and accumulates. Tumor cells usually show increased [^18^F]FDG uptake due to the overexpression of GLUT-1 transporter, increased intracellular hexokinase, and low FBP1 [51]. Since FBP1 activity might be suppressed only in some ccRCCs, [^18^F]-FDG presents a variable uptake in ccRCC lesions; from this biological mechanism, PET/CT with [^18^F] FDG is not currently recommended in ccRCC management. Among the included studies, two compared the performance of [^18^F]FDG and PSMA-targeted PET/CT and both reported superior performance of the latter [32,38]. This result may be explained by the different biological pathways underlying the uptake of the investigated tracers since PSMA-radioligands uptake relies on neoangiogenesis, which is upregulated in ccRCC by the VHL gene mutation and the subsequent HIFs accumulation, two rate-limiting factors in ccRCC development.

Carbonic anhydrase IX (CAIX), a cell membrane antigen, has been recently used as a target for molecular imaging in ccRCC patients [52], since its expression is upregulated in hypoxic conditions [53], induced in ccRCC by the loss of the VHL complex and the subsequent accumulation of HIFs. Two papers assessed the employment of the radiolabeled anti-CAIX antibody “girentuximab” labeled with [^89^Zr]Zr for the imaging of ccRCC [54,55]. They observed its ability to highlight ccRCC in a localized disease as well as in ccRCC metastases and to differentiate ccRCC from non-ccRCC lesions. In this context, no articles comparing PSMA-targeted PET to [^89^Zr]Zr-girentuximab were found in the literature search.

Since the introduction of carboxypeptidase type II as a possible target for molecular imaging and theragnostics, an increasing number of PSMA-targeting radiopharmaceuticals has been developed, including [^68^Ga]Ga-PSMA-11, [^18^F]F-DCFPyL, and [^18^F]F-PSMA-1007 [56,57]. Based on the available literature data, all the above-mentioned radiopharmaceuticals were employed for detecting ccRCC lesions [29,30,31,32,33,34,35,36,37,38,39,40,41,42]. Despite the use of [^18^F]-PSMA-targeting radiopharmaceuticals counts on several advantages such as reduced costs through large-scale production, higher-quality images through lower positron energy and a longer half-life, none of the included studies compared the diagnostic performances between different PSMA-targeting radiopharmaceuticals.

An intriguing issue underlying the use of PSMA-radioligands in ccRCC patients is based on the overexpression of carboxypeptidase type II by intratumoral neovascular endothelial cells. Since the current standard of care in patients with unresectable or relapsing metastatic ccRCC relies on antiangiogenic therapy with tyrosine kinase inhibitors and immunotherapy with checkpoint inhibitors [11], it is possible that PSMA-targeting radiopharmaceuticals might be able to assess and predict treatment response. Among the studies included in this systematic review, one compared PSMA-targeted PET/CT in the setting of the treatment response assessment to diagnostic CT and reported a concordance between the two examinations in a minority of patients [41]. Nevertheless, since no long-term follow-up data were reported, it was not possible to comment on the meaning of the discrepancies highlighted between the examinations and if they could improve patient management. In this setting, taking into account available evidence-based data, more studies concerning the predictive value or response assessment using this novel imaging method in ccRCC patients are needed.

Considering the recent introduction of PSMA-targeting radiopharmaceuticals labeled with β^−^ and α^+^ emitters, such as [^177^Lu]Lu-PSMA-617 and [^225^Ac]Ac-PSMA-617, and their outstanding results in metastatic castration-resistant prostate cancer patients [24,58], another interesting aspect of PSMA overexpression in ccRCC lesions is based upon the feasibility of a PSMA-targeted radioligand therapy. Nevertheless, the comprehensive literature search performed for this systematic review did not enhance any paper reporting data about RLT in ccRCC patients. In this context, the authors warrant clinical and pre-clinical investigations to assess the feasibility of a PSMA-targeted theragnostic approach in ccRCC patients and to evaluate its safety and efficacy.

Consistently with the studies included in this systematic review, Yin et al. reported a poor accuracy of PSMA-targeted PET/CT in patients with metastatic non-ccRCC, inferior to conventional imaging with CT and/or MRI [59]. These data may be explained by the different genetic pathways guiding the oncogenesis in the different types of RCC, since only ccRCC relies on the VHL mutation, which is strictly correlated with the accumulation of HIFs and the subsequent neoangiogenesis as a rate-limiting element in its development. Moreover, a recent prospective study has evaluated the role of PSMA-targeted PET/CT in the detection of gastrointestinal and pancreatic cancers (with the latter showing similar biologic characteristics to papillary RCC). The authors reported that the low PSMA tumor expression and the high physiological uptake in organs/background hampered the clear distinction of the tumor tissue and, as a result, [^18^F]FDG PET/CT was superior in detecting gastrointestinal and pancreatic tumors [60].

None of the studies included in this systematic review addressed the cost-effectiveness of PET with PSMA-targeting radiopharmaceuticals in managing ccRCC patients if it was actually introduced in clinical practice as a standard of care. Since, to date, this examination is employed in a research setting, cost-effectiveness studies are needed to strengthen and refine the role of this imaging technique in ccRCC.

Two previously published reviews explored the potential role of PSMA-targeted PET/CT in RCC patients [61,62]. The current systematic review had the purpose of providing an updated literature search using PRISMA guidelines as milestone in its editing, requiring a strict methodology, appropriate criteria for the literature search as well as for the quality assessment; moreover, case reports were excluded from the analysis since they are a possible source of bias. As a result, more data were provided in the results and discussion sections.

Concerning the limitations and biases of this systematic review, a limited number of prospective studies was available and most of them enrolled a restricted number of patients. Furthermore, we clearly demonstrated a significant heterogeneity among the included studies about the study methodology, patient characteristics, index test characteristics, and reference standard or comparison used. Prospective multicentric studies are needed to strengthen the role of PSMA-targeting PET/CT in ccRCC.

## 5. Conclusions

The qualitative data provided by this systematic review enhanced the promising role of PSMA-targeted PET/CT in patients with ccRCC. Nevertheless, more studies are warranted to overpower these findings and deepen how PET imaging with PSMA-targeting radiopharmaceuticals could integrate with conventional imaging in different clinical settings.

## Figures and Tables

**Figure 1 cancers-15-00355-f001:**
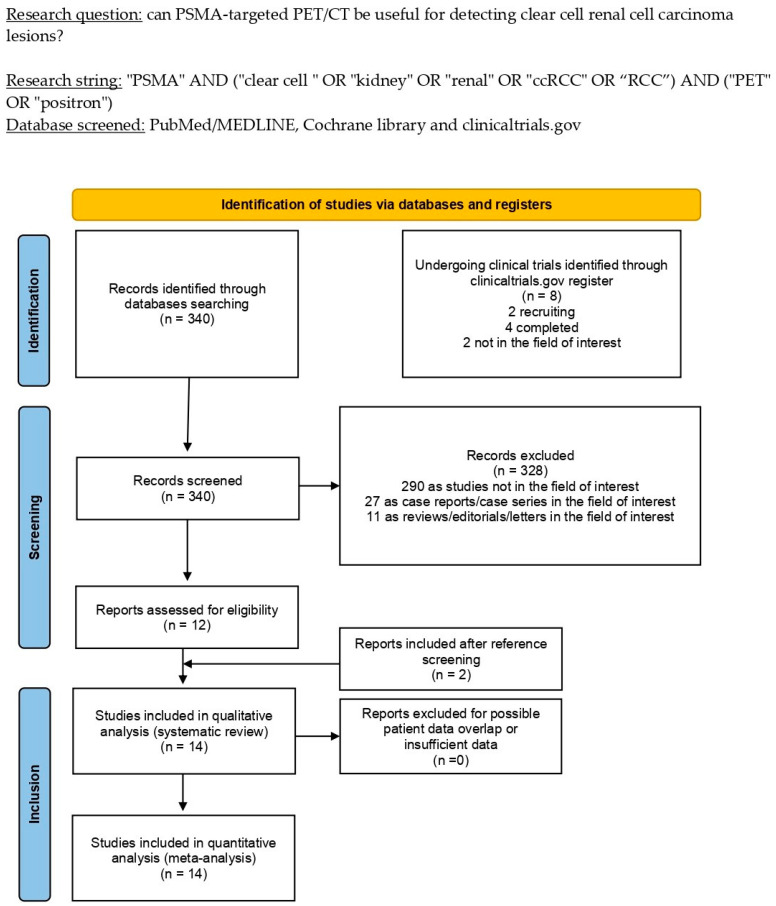
Comprehensive overview of the study selection process for the systematic review.

**Figure 2 cancers-15-00355-f002:**
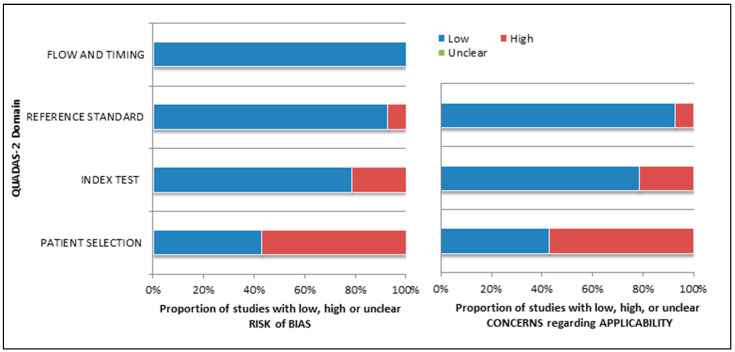
Summary of quality assessment according to the QUADAS-2 tool. The authors classified the papers included in the systematic review as a high risk or low risk of bias or applicability concerns for distinct domains listed in the ordinate axis. In contrast, the abscissa axis shows the percentage of studies. The graph indicates that about 60% of studies suffered a high risk of bias in “patient selection”, whereas a poor risk of bias was observed in “reference standard”, “index test”, and “flow and timing” domains.

**Table 1 cancers-15-00355-t001:** General study information.

Authors [Ref.]	Year	Country	Study Design/Number of Involved Centers	Funding Sources
Rhee et al. [29]	2016	Australia	Prospective/monocentric	None declared
Meyer et al. [30]	2019	U.S.A.	Prospective/monocentric	None declared
Raveenthiran [31]	2019	Australia	Retrospective/monocentric	None declared
Liu et al. [32]	2020	China	Retrospective/monocentric	None declared
Gao et al. [33]	2020	China	Retrospective/monocentric	None declared
Mittlmeier et al. [41]	2020	Germany	Retrospective/monocentric	None declared
Golan et al. [34]	2021	Israel	Prospective/monocentric	None declared
Gühne et al. [35]	2021	Germany	Retrospective/monocentric	None declared
Gao et al. [36]	2022	China	Retrospective/monocentric	None declared
Meng et al. [37]	2022	China	Retrospective/monocentric	None declared
Tariq et al. [38]	2022	Australia	Retrospective/multicentric	None declared
Tariq et al. [39]	2022	Australia	Retrospective/multicentric	None declared
Udovicich et al. [40]	2022	Australia	Retrospective/monocentric	None declared
Li et al. [42]	2022	China	Retrospective/monocentric	National Natural Science Foundation of China

**Table 2 cancers-15-00355-t002:** Patient key characteristics and clinical settings.

Authors [Ref.]	Sample Size (No. ccRCC Patients/All Included Patients)	Mean/Median Age (Years)	Gender (Male %)	Clinical Setting (No. Patients)	WHO/ISUP Grade (No. Lesions)	PSMA Staining Analysis	Comparative Imaging
Rhee et al. [29]	8/10	Median: 57	100%	10 staging	n.a.	n.a.	CT
Meyer et al. [30]	14/14	Median: 59	64.3%	14 staging	n.a.	n.a.	CT or MRI of abdomen + CT of the chest
Raveenthiran et al. [31]	28/38	Median: 64	25.7%	16 staging 22 restaging	n.a.	n.a.	CT
Liu et al. [32]	15/15	Mean: 57.5	86.6%	15 restaging	n.a.	n.a.	[^18^F]FDG PET/CT
Gao et al. [33]	36/36	Median: 61	58.3%	36 staging	9 grade 1 12 grade 2 9 grade 3 6 grade 4	Yes	No comparative exam
Mittlmeier et al. [41]	8/11	Mean: 59.6	72.7%	11 staging and restaging after treatment	n.a.	n.a.	CT
Golan et al. [34]	18/27	Median: 66	70%	27 staging	14 grade 1–2 4 grade 3	Yes	No comparative exam
Gühne et al. [35]	9/9	Range: 52–80	88%	9 Restaging	1 grade 1 6 grade 2 3 grade 3 1 grade 4	Yes	CT
Gao et al. [36]	37/48	Median: 59	60.4%	48 staging	10 grade 1 13 Grade 2 11 grade 3 7 grade 4	Yes	No comparative exam
Meng et al. [37]	40/53	Median: 56	60.4	53 staging	10 grade 1 17 grade 2 15 grade 3 3 grade 4	n.a.	No comparative exam
Tariq et al. [38]	10/11	Mean: 65.5	64%	4 staging 7 restaging	n.a.	n.a.	[^18^F]FDG PET/CT; CT
Tariq et al. [39]	14/14	Median: 61	64.3%	12 staging 2 restaging	3 grade 2 5 grade 3 5 grade 4 1 ungraded	n.a.	MRI
Udovicich et al. [40]	54/61	Mean: 65	56%	61 restaging	n.a.	n.a.	[^18^F]FDG PET/CT; CT
Li et al. [42]	40/50	Median age: 55	78%	50 staging	n.a.	Yes	CT or MRI

Legend: CT: computed tomography; FDG: fluorodeoxyglucose; ISUP: International Society of Urological Pathology; MRI: magnetic resonance imaging; n.a.: not available; PET: positron emission tomography; PSMA: prostate-specific membrane antigen; WHO: World Health Organization.

**Table 3 cancers-15-00355-t003:** Index test key characteristics.

Authors [Ref.]	Tracer	Hybrid Imaging	Tomograph	Administered Activity	Uptake Time (Minutes)	Image Analysis
Rhee et al. [29]	[^68^Ga]Ga-PSMA-11	PET/CT	Biograph mCT FLOW (Siemens ^®^)	150 MBq	60	Qualitative and semiquantative (SUV_max_)
Meyer et al. [30]	[^18^F]F-DCFPyL	PET/CT	n.a.	333 MBq	60	Qualitative and semiquantative (SUV_max_)
Raveenthiran et al. [31]	[^68^Ga]Ga-PSMA-11	PET/CT	Biograph mCT (Siemens ^®^)	150 MBq	45–60	Qualitative and semiquantitative (SUV_max_)
Liu et al. [32]	[^18^F]F-DCFPyL	PET/CT	Biograph 64 (Siemens ^®^)	0.15 Ci/kg	60	Qualitative and semiquantitative (SUV_max,_ TBR)
Gao et al. [33]	[^68^Ga]Ga-PSMA-11	PET/CT	uMI 780 (United Imaging Healthcare ^®^)	n.a.	45	Qualitative and semiquantitative (SUV_max_, SUV_mean_, SUV_peak_)
Mittlmeier et al. [41]	[^18^F]F-PSMA-1007	PET/CT	Biograph mCT (Siemens ^®^); Biograph 64 (Siemens ^®^)	217–268	60	Qualitative and semiquantitative (SUV_mean_, SUL)
Golan et al. [34]	[^68^Ga]Ga-PSMA-11	Dynamic PET/CT	Discovery 710 (GE healthcare ^®^)	75–150 MBq	0	Qualitative, semiquantitative (SUV_max_, SUV_mean_, TBR) and kinetic analysis
Gühne et al. [35]	[^68^Ga]Ga-PSMA-11	PET/CT	Biograph mCT 40 (Siemens ^®^)	221–272 MBq	74–103	Qualitative and semiquantitative (SUV_max_, SUV_mean_, TBR)
Gao et al. [36]	[^68^Ga]Ga-PSMA-11	PET/CT	uMI 780 (United Imaging Healthcare ^®^)	n.a.	45	Qualitative and semiquantitative (SUV_max_)
Meng et al. [37]	[^68^Ga]Ga-PSMA-11	PET/CT	uMI 780 (United Imaging Healthcare ^®^)	n.a.	45	Qualitative and semiquantitative (SUV_max_)
Tariq et al. [38]	[^68^Ga]Ga-PSMA-11; [^18^F]F-PSMA-1007	PET/CT	Biograph mCT (Siemens ^®^); Ingenuity TF (Philips ^®^); Discovery MI DR (GE ^®^)	124–168 MBq for [^68^Ga]Ga-PSMA-11; 224–244 MBq for [^18^F]F-PSMA-1007	45–63 for [^68^Ga]Ga-PSMA-11; 120–130 for [^18^F]F-PSMA-1007	Qualitative and semiquantitative (SUV_max_)
Tariq et al. [39]	[^68^Ga]Ga-PSMA-11; [^18^F]F-PSMA-1007	PET/CT	Biograph mCT (Siemens ^®^); Ingenuity TF (Philips ^®^); Discovery MI DR (GE ^®^)	121–267 MBq for [^68^Ga]Ga-PSMA-11; 247–260 MBq for [^18^F]F-PSMA-1007	41–94 for [^68^Ga]Ga-PSMA-11; 117–142 for [^18^F]F-PSMA-1007	Qualitative and semiquantitative (SUV_max_)
Udovicich et al. [40]	[^68^Ga]Ga-PSMA-11; [^18^F]F-DCFPyL	PET/CT	Discovery 690 (GE ^®^); Discovery 710 (GE ^®^)	2.6 MBq/kg for [^68^Ga]Ga-PSMA-11; 3.6 MBq/kg for [^18^F]F-DCFPyL	60 for [^68^Ga]Ga-PSMA-11; 120 for [^18^F]F-DCFPyL	Qualitative and semiquantitative (SUV_max_, MTV, TLP)
Li et al. [42]	[^68^Ga]Ga-PSMA-11	PET/CT	Biograph mCT.X (Siemens ^®^)	0.05 mCi/kg	60–90	Qualitative and semiquantitative (SUV_max_, TBR)

Legend: DCFPyL: piflufolastat; MTV: metabolic tumor volume; n.a.: not available; PET/CT: positron emission tomography/computed tomography; PSMA: prostate-specific membrane antigen; SUL: standard uptake value corrected for lean body mass; SUV: standard uptake value; TBR: target-to-background ratio; TLP: total lesion PSMA.

**Table 4 cancers-15-00355-t004:** Outcomes of the included studies.

Authors [Ref.]	Primitive Lesion SUV_max_	Metastatic Lesions SUV_max_	Correlation of Uptake with Histology or IHC	Detection Rate	Change of Management (No. Patients)
Rhee et al. [29]	Mean: 18.0	Mean: 19.5	n.a.	Per-lesion: 100%	2 (20%)
Meyer et al. [30]	Median: 9.6	Median: 2.7	n.a.	Per-patient: 92.8% Per-lesion: 88.9%	3 (21.4%)
Raveenthiran et al. [31]	n.a.	n.a.	n.a.	Per-lesion: 80.5%	Staging: 7 (43.8%) Restaging: 9 (40.9%)
Liu et al. [32]	n.a.	Soft tissue mean: 6.9 Bone mean: 8.2	n.a.	Per-patient: 100% * Per-lesion: 100% *	n.a.
Gao et al. [33]	Mean: 17.78	n.a.	SUV_max_ differentiates WHO/ISUP grade and adverse pathology	n.a.	n.a.
Mittlmeier et al. [41]	n.a.	n.a.	n.a.	n.a.	9 (81.8%)
Golan et al. [34]	Median: 9.4	n.a.	Cytoplasmatic PSMA staining was associated with the washout coefficient.	n.a.	n.a.
Gühne et al. [35]	n.a.	Median: 3.1	No correlation between uptake and PSMA staining in IHC analysis	n.a.	n.a.
Gao et al. [36]	n.a.	n.a.	SUV_max_ differentiates VEGFR-2 expression, PDGFR-β expression, and VEGFR-2 and PDGFR-β coexpression	n.a.	n.a.
Meng et al. [37]	n.a.	n.a.	SUV_max_ correlates with high HIF-2α expression	n.a.	n.a.
Tariq et al. [38]	Median: 3.2	Median: 8.0	n.a.	n.a.	3 (27%)
Tariq et al. [39]	Mean: 25.3	n.a.	n.a.	n.a.	3 (30%)
Udovicich et al. [40]	n.a.	15.0	n.a.	Per-patient: 84%	30 (49%)
Li et al. [42]	Median: 18	Median: 3.7–9.6 (based on location)	SUV_max_ values are related to pathologic subtypes and PSMA staining scores	Per-lesion: 93.6%	4 (12.9%)

Legend: HIF-2α: hypoxia induced factor 2α; IHC: immunohistochemistry; ISUP: International Society of Urological Pathology; PDGFR-β: platelet-derived growth factor receptor β; PSMA: prostate-specific membrane antigen; SUV_max_: maximum standard uptake value; VEGFR-2: vascular endothelial growth factor receptor 2; WHO: World Health Organization. * = only [^18^F]FDG PET/CT used as reference standard to calculate detection rate values.

## Data Availability

The data presented in this study are available on request from the corresponding author.

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
