# Peer review of "The Emerging Role of PET/CT with PSMA-Targeting Radiopharmaceuticals in Clear Cell Renal Cancer: An Updated Systematic Review"

_cancers, 2023, doi:10.3390/cancers15020355_

Round 1

Reviewer 1 Report

This review about PET/CT with PSMA in ccRCC is very complete.A new imagery with new radio pharmaceutical tracers.Apart from the diagnosis , the real interest of this imaging is the possibility to change the treatment or to predict  the response. However, the results aren’t convincing: 13 to 43% to change the management.For the urologist , the presence of tumor thrombi is essential to operate and RMI is better.

We need mora data and more prospective  clinical trials to assess the predictive value of response 

What is the cost of this imaging compared to conventional imaging ?Can be accessible to all centres ?

1/ The main question is the role of PET-PSMA imaging and renal cancer. 2/ I think , it is a very interested question because we don’t have the best tool to evaluate with precision the stage of clear cell renal cell carcinoma cancer( ccRCC). At the moment , we used always the conventional imaging. 3/ PET/CT with PSMA is a new radio pharmaceutical.The carboxypeptidase type II is expressed by prostate cancer and by ccRCC.This review brings a focused vision of this technique in the field of ccRCC. 4 and 7 /Perhaps , they need to explain better the quality assessment (QUADAS-2) and the figure 2 5/Yes, because the open the door for new prospective trials. 6/Yes , the references are appropriate and updated.

Author Response

This review about PET/CT with PSMA in ccRCC is very complete. A new imagery with new radio pharmaceutical tracers. Apart from the diagnosis , the real interest of this imaging is the possibility to change the treatment or to predict the response. However, the results aren’t convincing: 13 to 43% to change the management. For the urologist , the presence of tumor thrombi is essential to operate and RMI is better.

Reply: We thank the reviewer for the precious comments on our manuscript. We also added some comments of the reviewer in the revised manuscript.

We need more data and more prospective clinical trials to assess the predictive value of response.

Reply: According to the reviewer’s suggestion we have added in the discussion that, taking into account available evidence-based data, we need more data on the predictive value or response assessment by using this novel imaging method in ccRCC

What is the cost of this imaging compared to conventional imaging? Can be accessible to all centres?

Reply: We have added in the revised manuscript that the cost of this novel hybrid imaging method is higher compared to conventional imaging methods. Furthermore, this method is currently used only in a research setting. We have specified in the revised manuscript that cost-effectiveness studies are needed to strengthen the role of PSMA-targeted PET/CT in ccRCC.

1/ The main question is the role of PET-PSMA imaging and renal cancer. 2/ I think , it is a very interested question because we don’t have the best tool to evaluate with precision the stage of clear cell renal cell carcinoma cancer( ccRCC). At the moment , we used always the conventional imaging. 3/ PET/CT with PSMA is a new radio pharmaceutical.The carboxypeptidase type II is expressed by prostate cancer and by ccRCC.This review brings a focused vision of this technique in the field of ccRCC. 4 and 7 /Perhaps , they need to explain better the quality assessment (QUADAS-2) and the figure 2 5/Yes, because the open the door for new prospective trials. 6/Yes , the references are appropriate and updated.

Reply: We thank the reviewer for the precious comments. WE explain better the QUADAS-2 tool and the figure 2.

Reviewer 2 Report

I congradulate the aothors on the courage for publishing the paper.

I suggest to the outhors to contemplate on the following:

1. since the male population is in the paper PSA and mpMRI of the prostate should be dane to exclude prostate cancer.

2. if it is possible androgen receptors (AR) on renal tissue and metastases positive to PSMA-PET should be analyzed

3. to investigate AR on panceratic cancer  which are positive to PSMA-PET (pancreatic cancer has simular biological characteristics as the mCRPC)

Author Response

  1. since the male population is in the paper PSA and mpMRI of the prostate should be dane to exclude prostate cancer.

Reply: we specified in the revised manuscript that PSMA-targeted PET/CT does not substitute PSA and MRI to detect prostate cancer

  1. if it is possible androgen receptors (AR) on renal tissue and metastases positive to PSMA-PET should be analyzed

Reply: unfortunately the included articles do not report data on androgen receptors on ccRCC primary tumors, relapse and metastases, therefore we are not able to add this information.

  1. to investigate AR on panceratic cancer which are positive to PSMA-PET (pancreatic cancer has simular biological characteristics as the mCRPC)

Reply: as our systematic review article is focused in ccRCC we have not provided data on pancreatic cancer at PSMA-targeted PET/CT or androgen receptor status of pancreatic cancer. However, we have added in the discussion that a recent prospective study has evaluate the role of PSMA-targeted PET/CT in the detection of gastrointestinal and pancreatic cancers. Low PSMA tumor expression and high physiological uptake in organs/background hamper the clear distinction of the tumor. As a result, [18F]FDG PET/CT was superior in detecting gastrointestinal and pancreatic cancers.

We have added this study to the reference list: Vuijk FA, Kleiburg F, Noortman WA, Heijmen L, Feshtali Shahbazi S, van Velden FHP, Baart VM, Bhairosingh SS, Windhorst BD, Hawinkels LJAC, Dibbets-Schneider P, Bouwman N, Crobach SALP, Fariña-Sarasqueta A, Marinelli AWKS, Oprea-Lager DE, Swijnenburg RJ, Smit F, Vahrmeijer AL, de Geus-Oei LF, Hilling DE, Slingerland M. Prostate-Specific Membrane Antigen Targeted Pet/CT Imaging in Patients with Colon, Gastric and Pancreatic Cancer. Cancers (Basel). 2022 Dec 15;14(24):6209. doi: 10.3390/cancers14246209.